# Human Papillomavirus: Possible Mechanisms of Damage in Sinonasal Inverted Papilloma

**DOI:** 10.3390/ijms27010245

**Published:** 2025-12-25

**Authors:** Ana Karla Guzmán-Romero, Rebeca Pérez Cabeza de Vaca, Giovani Visoso-Carvajal, Moises Lopez-Gonzalez, Carmen Selene García-Romero, Jazmín García-Machorro

**Affiliations:** 1Servicio de Otorrinolaringología, Centro Médico Nacional “20 de Noviembre”, Instituto de Seguridad y Servicios Sociales de los Trabajadores del Estado, Mexico City 03100, Mexico; drakguzman.orl@gmail.com; 2Laboratorio de Medicina de Conservación, Escuela Superior de Medicina, Instituto Politécnico Nacional, Mexico City 11340, Mexico; carvajalgv@gmail.com; 3Centro de Ciencias de la Complejidad UNAM, Mexico City 04510, Mexico; rebeca.perez@c3.unam.mx; 4Department of Microbiology, Immunology, and Molecular Genetics, David Geffen School of Medicine, University of California, Los Angeles, CA 90024, USA; moiseslopez@mednet.ucla.edu; 5Laboratorio de Biología Molecular y Virología, Instituto Nacional de Perinatología. Secretaría de Salud, Mexico City 11000, Mexico

**Keywords:** finger-like projection, nasosinusal inverted papilloma, pathogenesis, human papillomavirus, development, progression, recurrence

## Abstract

Sinonasal inverted papilloma (SNIP) is a benign neoplasm derived from the Schneiderian membrane and the endoderm of the ciliated respiratory epithelium of the nasal cavity and paranasal sinuses. SNIP is uncommon and typically found between the fourth and seventh decades of life, with men being more frequently affected. The medical significance of this benign neoplasm lies in its potential to cause local tissue destruction, tendency toward malignancy, and high recurrence rate. This study describes the histology of the nasosinusal mucosa and histological characteristics of SNIP, as well as its clinical manifestations and treatment. We also describe findings in the development of SNIP such as chronic inflammation and environmental factors. Additionally, we describe the association between cases positive for the human papillomavirus (HPV) and progression to malignancy and recurrence. This analysis includes the tumor microenvironment, encompassing the infiltration of immune cells such as CD4+ and CD8+ lymphocytes, macrophage polarization, and increases in certain metalloproteinases (MMP-2 and MMP-9). Finally, we address epigenetic alterations associated with HPV infection.

## 1. Introduction

Sinonasal inverted papilloma (SNIP) is a benign neoplasm derived from Schneider’s membrane (cells in contact with the periosteum of the maxillary sinus, which form a pseudostratified respiratory epithelium, including mucus-secreting goblet cells and numerous serous glands) and the endoderm of the ciliated respiratory epithelium of the nasal cavity and paranasal sinuses. SNIP belongs to a group of papillomas known as Schneiderian, which present as unilateral sinonasal lesions (5% of cases are bilateral). According to their histopathological characteristics, Schneiderian papillomas are divided into 3 types: (1) oncocytic or columnar characterized by the columnar epithelium, whose implantation site is located in the lateral nasal wall; (2) fungiform or exophytic with the presence of squamous epithelium, originating in the nasal septum; (3) sinonasal inverted papilloma formed by the transitional epithelium with downward growth towards the underlying stroma, derived from the lateral nasal wall, middle meatus, or paranasal sinuses [1,2,3,4].

SNIP is histopathologically characterized by inverted growth of the epithelium into the stroma with a finger-like projection, where the basement membrane completely separates the epithelial component from the underlying connective tissue stroma; increased thickness of the epithelium, which may lead to squamous metaplasia (i.e., the epithelial tissue transforms into a squamous epithelium); the presence of inflammatory cell infiltrates such as neutrophils, macrophages, and lymphocytes in the epithelium and stroma of the tissue; a small SNIP initiation site compared to the tumor body [5].

Worldwide, SNIP is the second most common benign tumor of the nose and paranasal sinuses, after osteoma. It represents 0.4–4.7% of all nasal tumors, with an incidence of 0.74–2.3 new cases per 100,000 inhabitants and is more common in men with a ratio of 2.3:1, with peak incidence between the fifth and sixth decades of life [6].

The lesions are located in the nasal cavity and paranasal sinuses. The most frequent site of origin or implantation is the lateral nasal wall in up to 42% of cases; SNIP also originates in the maxillary sinus (26%) and less frequently in the sphenoid, frontal, and ethmoid sinuses or the nasal septum [7]. In some cases, multiple implantation sites are found —a characteristic associated with a higher recurrence rate [8].

The histopathological characteristics and frequency of SNIP have been described in multiple reviews. However, the role of human papillomavirus (HPV) in these lesions has been under-researched. Indeed, there is conflicting information regarding the relationship between high risk (HR-HPV) and low risk (LR-HPV) [9]. In this review, we describe the possible molecular mechanisms that may contribute to the development, recurrence, and transformation of SNIP and their relationship with HPV.

## 2. Sinonasal Histology

The respiratory system constantly filters the external environment during respiration. The airways must maintain the capacity to eliminate inhaled pathogens, allergens, and debris to maintain homeostasis and prevent inflammation. The respiratory system is subdivided into a conducting portion and a respiratory portion. The nasal vestibule, which connects to the exterior, extends from the front of the nose to the posterior two-thirds. The nasal vestibule is the only part of the nasal cavity that contains keratinocytes and is lined with a keratinized stratified squamous epithelium, similar to the epidermis of the face. The nasal vestibule also contains glands and vibrissae, which constitute a protective barrier [10].

Where the nasal vestibule ends (posteriorly), the stratified squamous epithelium thins and transitions to the columnar epithelium, known as the respiratory mucosa. The respiratory mucosa, which lines the posterior nasosinusal cavity and turbinates, is composed primarily of pseudostratified ciliated columnar epithelium (almost 80%), with specialized cells such as goblet cells, brush cells, and basal cells [10]. The respiratory epithelium contains about 200 cilia per cell [11]. Beneath the epithelium lies the lamina propria, which is rich in blood vessels and contains serous and mucous glands that produce mucus secretion [12]. The mucosae of the nose and paranasal sinuses play a protective role in the upper respiratory tract through mucociliary clearance, the mechanism by which mucus is produced and transported, allowing for the removal of toxic molecules trapped within it [13] (see Figure 1A,B).

In general, the respiratory mucosa is mainly composed of the following cells:

Basal cells: These cells are attached to the basal membrane via receptors (integrins) and hemidesmosomes. These cells act as progenitor cells capable of differentiating into goblet cells or ciliated columnar cells, which provide protection against the abrasive sinonasal environment and possibly help to anchor overlying cells [13,14].

Brush cells: These cells possess short, blunt microvilli and receptors that detect substances, modulating inflammation and the immune response, although their exact function is still under investigation [15].

Goblet cells: These cells are distributed among the columnar cells and contain microgranules; they synthesize, store, and secrete mucin, a glycoprotein essential for the viscosity and elasticity of mucus.

Non-ciliated columnar cells: These cells play an important role in airway fluid homeostasis, as they secrete proteins, enzymes, and pro-inflammatory cytokines, such as interleukin-17C, to detoxify inhaled harmful substances.

Ciliated columnar cells: These cells constitute almost 80% of the total cell population; their cilia exhibit a coordinated upward and outward movement for the expulsion of foreign particles and pathogens [12,13,14].

All columnar cells, both ciliated and non-ciliated, have hundreds of immobile microvilli on their surfaces, which correspond to actin filaments 1 to 2 µm in length, covered by the cell membrane. This factor increases the total surface area of the columnar cells, which contributes to the production, secretion, and sensitivity of mucus in the nasosinusal mucosa.

The basement membrane is a thin, non-cellular layer of the specialized extracellular matrix, which functions as a barrier that physically separates the epithelium from the underlying connective tissue, which contains capillaries. The lamina propria contains nerve structures, blood vessels, and glands [14].

Seromucinous glands produce serous and mucinous secretions and are found throughout the cavity. Intraepithelial glands consist of goblet cells arranged around a lumen. The parasympathetic fibers of the lamina propria directly stimulate glandular secretions. Sympathetic fibers of the lamina propria, on the other hand, act to vasoconstrict and decongest the mucosa [13].

### 2.1. Histological Characteristics of Sinonasal Inverted Papilloma

A distinctive histological feature of SNIP is invagination of the surface epithelium into the underlying stroma, hence the term “inverted” [3]. Cellular structures composed of macrophages, intraepithelial microcysts, cellular debris, and mucinous material are also observed [6]. Mitotic activity may be present in the basal and parabasal layers. The stromal component ranges from myxoid edematous to fibrous, frequently containing chronic inflammatory cells and variable vascularization [16]. The neoplastic epithelium is typically thickened, inverted into the underlying connective tissue, with an intact basement membrane, and is generally composed of the columnar or ciliated respiratory epithelium with varying degrees of squamous differentiation. The amount of inflammatory infiltrate is usually associated with less clinically aggressive lesions [4]. The progression to malignancy of SNIP is described below.

Hyperplasia: Defined as a >7-fold increase in the number of epithelial layers.

Dysplasia: Cytological and, occasionally, architectural anomalies, the quantification of which classifies the grade of dysplasia and informs the prognosis.

Low-grade dysplasia: Reactive anomalies presenting with minimal to no cellular atypia, infrequent mitoses, and mild dyskeratosis.

High-grade dysplasia: Encompasses both moderate and severe dysplasia, architecturally defined by disordered stratification, while cytology reveals hyperchromatic nuclei, an elevated nuclear-to-cytoplasmic ratio, atypical mitoses beyond the basal layer, and dyskeratosis with eosinophilic cells.

Carcinoma in situ (CIS): Confined to severe and infrequent architectural disturbances, significant cellular and nuclear atypia, and multiple atypical mitoses.

Microinvasive carcinoma: Characterized by a breach of the basement membrane and invasion of the lamina propria [17].

### 2.2. Medical Importance

SNIP is generally considered a benign tumor. However, it can occasionally exhibit locally aggressive behavior, with a high rate of recurrence and malignancy. SNIP associated with malignancy is classified into two types: synchronous and metachronous. Synchronous SNIP refers to the neoplasm’s coexistence with carcinoma at the time of initial diagnosis. In these cases, the carcinoma may originate from the papilloma or present as a separate lesion. The literature reports an average synchronous carcinoma incidence of up to 7% of cases, among which squamous cell carcinoma (SCC) is the most common histological type [8].

Metachronous carcinomas arise in the same anatomical location where a SNIP previously developed. The mean time to develop a metachronous carcinoma is 52 months (range: 6 to 180 months), and the estimated malignant potential for recurrence is up to 11% [18]. Metachronous tumors have a relatively better prognosis than synchronous tumors and de novo SCC (5-year overall survival: 73.1%, 54.5%, and 55.4%, respectively) [19]. This difference in prognosis occurs because de novo SCC is considered more aggressive, as it is not related to a previously treated precancerous lesion. This factor differs from carcinoma ex-SNIP, which arises as a SNIP and undergoes resection treatment, only to recur and potentially exhibit an increase in immune response cells to eliminate the tumor or even keratinized cells that indicate the need for immediate attention.

### 2.3. Clinical Manifestations

Despite being benign tumors, SNIP present three distinctive characteristics: they are locally aggressive lesions, have a high recurrence rate, and a certain percentage is associated with malignancy. Recurrence depends on several factors: the implantation site, lesion size, and the type of surgical technique used for resection. However, a recurrence rate of 2% to 27% has been reported [20]. On the other hand, the literature reports a very wide range of cases that develop malignancy (between 0% and 53%) [21]. Most authors agree that malignancy is observed in approximately 5% to 15% of cases [22,23]. SCC is the predominant malignant lesion present. Other reported tumors include adenocarcinoma, mucoepidermoid carcinoma, and verrucous carcinoma [8].

Clinical manifestations are usually nonspecific, primarily unilateral nasal obstruction, accompanied in up to 50% of cases by anterior and posterior rhinorrhea (20.8%) [7], followed by hyposmia, headache, or central facial pain and, in some cases, epistaxis [24]. Diagnosis is usually made in advanced stages, as the lesions are initially asymptomatic. During a physical examination, nasal endoscopy reveals exophytic, reddish-gray, papillomatous lesions, typically originating from the lateral nasal wall [8].

### 2.4. Etiology and Mechanisms of Damage

A specific etiology remains undetermined, and the risk factors are controversial. The mechanisms for the formation and growth of finger-like projections that disrupt and insert into the tissue stroma as a characteristic part of the pathogenesis of SNIP are not well understood. In this review, we focus on the factors that generate chronic inflammation and the mechanisms associated with the human papillomavirus (HPV).

#### 2.4.1. Chronic Inflammation

According to several studies, chronic inflammation has been associated with the development of SNIP, which most frequently originates in the lateral nasal wall or maxillary sinus, within the nasosinusal epithelium of Schneider, where chronic inflammatory changes are common [4,25].

In a study of 50 Chinese patients with unilateral SNIP, tissue was obtained and compared with inferior turbinate mucosal biopsies from 17 healthy subjects. Histological patterns, epithelial remodeling, and inflammatory cell infiltration were evaluated. Moderate and severe remodeling with the infiltration of macrophages; eosinophils; CD8+ T cells; and T-reg cells, predominantly neutrophils, was found in 49% of patient samples [16].

In another study supporting the high number of neutrophil infiltrates, nasal papilloma and inferior turbinate samples were obtained from patients with SNIP (*n* = 50) and from control subjects with a deviated septum (*n* = 15). IL-17+ cells were assessed via immunohistochemistry and flow cytometry. Higher positive expression was observed in the SNIP samples (in mononuclear cells and neutrophils) compared to the controls [26].

Additionally, a study involving 65 patients diagnosed with SNIP and 65 healthy controls of the same age and sex compared inflammatory markers in the blood, such as the neutrophil lymphocyte ratio (NLR), platelet lymphocyte ratio (PLR), red cell distribution width (RDW), mean platelet volume (MPV), and platelet distribution width (PDW). No significant differences were observed in NLR, PLR, RDW, MPV, and PDW values between the two groups (*p* > 0.05). However, when constructing a logistic regression analysis model to investigate the effects of inflammatory blood markers on patient group determination, the increase in NLR and decrease in PLR were found to be statistically significant factors (*p* = 0.008, *p* = 0.039, respectively) [27].

Recently, another study compared the histopathological patterns (degrees of epithelial remodeling and p63 and CK5 expression via immunohistochemistry) and inflammatory characteristics of SNIP with those of chronic rhinosinusitis with nasal polyps (CRSwNPs) or normal ethmoid sinus mucosa. In total, the study analyzed 58 tissue biopsies from 38 patients with SNIPs, 12 CRSwNPs, and 8 normal ethmoid sinus mucosae. Grade II remodeling was found in most SNIP samples (36.8%). Here, p63 and CK5 expression levels were significantly higher in the SNIP group than in the other two groups (*p* < 0.05 in both cases), as were levels of neutrophil and macrophage infiltration and the expression levels of the inflammatory cytokines interleukin-1β, interleukin-6, and tumor necrosis factor-α. Therefore, it was concluded that, in addition to inflammation, excessive remodeling also occurs in SNIP [28].

#### 2.4.2. Environmental Factors

The environmental factors that cause chronic inflammation and may contribute to the etiology of SNIP include occupational, environmental, and industrial exposure and smoking, which are related to recurrence and transformation.

To determine the risk factors associated with SNIP, a study conducted in China included 50 cases and compared them with 150 controls matched for sex and age. This study found that ten patients (20%) in the papilloma group experienced industrial exposure, which included the construction, textile, printing, paper, and electronics industries. Eight patients (5.3%) in the control group had industrial exposure in the construction, textile, and food processing industries. In the univariate analysis, industrial exposure was determined as a risk factor with a significant difference of *p* = 0.002 (odds ratio de 4.44, 95% CI = 1.64–1.97) [29].

Another case–control study conducted in Piedmont (a region of Italy dedicated to the metallurgical industry a century ago) analyzed questionnaires from 337 controls and 127 cases matched for age, social class, sex, and province of residence. The authors inquired about substances used—by the subjects themselves and by others working near them—such as arsenic, wood dust, leather dust, nickel compounds, chromium VI and its salts, polycyclic aromatic hydrocarbons, welding fumes, oil mists, formaldehyde, flour, cocoa powder, textile dusts, silica, coal dust, paint mists, strong acid mists, and organic solvent vapors. Exposure to organic solvents for more than five years was associated with a statistically significant increased risk (OR 2.55) [30].

In general, smoking is considered the most important risk factor in the development and recurrence of head and neck neoplasms. However, in the case of SNIP, smoking is not associated with development but, rather, with recurrence and malignant transformation of this type of tumor [31].

In a systematic review and meta-analysis, the risk factors for malignant transformation in SNIP were analyzed. The authors analyzed 1271 SNIP cases, with a carcinoma incidence of 230/1271 (18.1%), concluding that both smoking and HPV are risk factors for the malignant transformation of SNIP, with *p* values = 0.002 and *p* < 0.001, respectively [32].

#### 2.4.3. Association with the Human Papillomavirus

Recently, Human Papillomavirus (HPV) has been identified as an etiologic agent of nasal and oropharyngeal carcinomas. Patients with this association are younger and less associated with a history of tobacco and alcohol consumption. However, it was also recently shown that HPV infection has an independent role as a risk factor and is related to a different prognosis than that in patients without viral infection [32].

Papillomavirus encompasses a family of 150 virus types that can infect epithelial cells such as those of the skin, oral mucosa, and genitals. The different types of these viruses can be grouped according to the sequence of the nucleotides, defined as alpha or beta, and, in turn, can be divided into high-risk or low-risk groups [33,34].

Bosch et al. suggested a classification for the different known genotypes of HPV [35]. A total of 15 genotypes were classified into types with high oncogenic risk (HR-HPV) (16, 18, 31, 33, 35, 39, 45, 51, 52, 56, 58, 59, 68, 73, and 82); 3 with probable high oncogenic risk (26, 53, and 66); and 12 with low oncogenic risk (LR-HPV) (6, 11, 40, 42, 43, 44, 54, 61, 70, 72, 81, and CP6108). Currently, 15 types of HPV are considered HR-HPV because they produce precancerous lesions and cancerous lesions; the relevant types are 16, 18, 31, 33, 34, 35, 39, 45, 51, 52, 56, 58, 59, 66, 68, and 70. Low-risk lesions associated with benign genital warts and other non-malignant lesions, such as respiratory papillomatosis and oral focal epithelial hyperplasia, are types 6, 11, 42, 43, and 44 [36,37].

The HPV viral genome consists of double-stranded, circular, and covalently closed DNA with 8000 base pairs. The viral genome of this virus comprises three different regions. The E region, a gene that encodes early viral function, includes E1, E2, E4, E5, E6, and E7. L is a gene that encodes the viral capsid (L1 and L2) and the CSF region that is responsible for normal virus replication and gene expression control [38]. HPV relies on host cell replication machinery for genome replication because its genome does not encode its own DNA polymerase.

The virus follows a replication cycle closely linked to the differentiation of epithelial cells, like that which occurs in other mucous membranes. HPV binds to basal cells via integrin α6β4 and syndecan, which are expressed in cells following mucosal microlesions that create an inflammatory environment [39]. Furthermore, the expression of proteins such as neuronal cadherin (N-cadherin), vimentin, integrin, fibronectin, and matrix metalloproteinases (MMPs) has been described. These proteins cause aberrant differentiation of the nasal epithelium, altering its phenotypic characteristics and contributing to greater damage and predisposition to infections [11]. Once undifferentiated cells are infected, the viral genome is maintained as an episome (independent circular DNA) with a low copy number of approximately 50–100 per cell.

Subsequently, during basal cell division and detachment from the basal epithelium, the viral genome replicates in infected cells in sync with cellular DNA replication; the different proteins in the E region function differently. E2 forms a heterodimer with E1 to initiate viral replication [38]. E7 induces cell proliferation [38,40] by arresting the cell’s G1–2 phase, and this protein inhibits cyclin-dependent kinases (CDKs) and retinoblastoma protein [38]. E6 inactivates p53; therefore, E6 and E7 deregulate the cell cycle and allow amplification of the viral genome [38]. The cells then begin to differentiate and migrate to the surface, while the virus activates its productive phase. In the suprabasal layers, massive amplification of the genome occurs, reaching up to 1000 copies per cell. Subsequently, in the outermost layers of the sinonasal epithelium, the capsid proteins (L1 and L2) are expressed to encapsulate the new viral genomes. The new viral particles are released into the environment when mature cells are shed from the mucosal surface [39].

The prevalence of HPV in the USA has been reported in 65% of cases with oropharyngeal cancer subtype 16, with the most frequent being head and neck cancer. Subtype 18 and 33 infections have also been reported. In Mexico, there are few reports of the incidence of HPV 16 in head and neck tumors. Gallegos et al., in a retrospective study, found that among 118 cases of head and neck tumors, 70% were positive for HPV 16, with no direct relation to smoking, with the larynx, gums, and tongue corresponding to the highest levels of positive results [32].

##### Association with the Development of SNIP

The molecular mechanisms through which HPV induces head and neck cancer are not well understood. Even the causal or developmental association of SNIP with HR-HPV and LR-HPV is controversial. In a retrospective study of tissue samples from patients diagnosed with SNIP and consecutively treated with endoscopic resection, the presence of the HPV genome was sought using multiplex genotyping based on polymerase chain reaction (PCR). HPV DNA sequences were identified in 34 of 55 HPV cases (61.8%), with a higher prevalence of HR-HPV genotypes (19 cases [55.9%]) than LR-HPV genotypes (15 cases [44.1%]). HPV16 (84.2%) and HPV54 (53.3%) were prevalent in HR-HPV and LR-HPV cases, respectively [41].

In a systematic review of HPV types among patients with benign SNIP, SNIP with dysplasia, and malignant SNIP (SCC ex-SNIP), a ratio of 2.8:1 was determined for LR-HPV and HR-HPV cases. Patients with benign SNIP were 4.8 times more likely to have LR-HPV than patients with HR-HPV, and patients with SCC ex-SNIP were 2.4 times more likely to have HR-HPV. The authors suggested that LR-HPV may induce SNIP formation and then become lost to detection as infected cells are shed [42]. This hit-and-run theory stipulates that HPV and other viruses can induce mutations and damage to the genetic structure, enabling the formation of tumor cells [9,43].

LR-HPV has also been associated with the development of SNIP. For example, a lentivirus model with high expression of HPV11E6/E7 was stably transfected into Human Nasal Epithelial Cells (HNEpC), and the KDM4A (histone lysine-specific demethylase 4a) gene was knocked out via CRISPR/Cas9 gene editing technique mediated by electroporation. The study groups were HNEpC as the negative control group, HPV11E6/E7-HNEpC as the positive control group, and koKDM4AHPV11E6/E7-HNEpC as the experimental group; these groups were used to detect changes in cell proliferation and migration. The results showed that overexpression of HPV11E6/E7 promoted the proliferation and migration of HNEpC cells, and knockout of the KDM4A gene significantly inhibited the proliferation of HPV11E6/E7-positive cells. Additionally, in vivo experiments were performed using an HPV11-positive subcutaneous tumor model of SNIP mice to verify the effects of the KDM4A gene knockout. In total, 2 × 10^6^ cells from the study groups were injected into female BALB/c nude mice. The weights of the mice and the volumes of the tumors were measured for 21 days. Subsequently, the tumor tissue was extracted, and the expression levels of HPV11E6, HPV11E7, and KDM4A were measured via qRT-PCR. M1 and M2 macrophage infiltration was also assessed with immunofluorescence. Overexpression of HPV11E6/E7 was found to recruit more macrophages to the tumor and induce their transformation into the M1 type [44].

Another theory is that the HPV E5 protein induces an increase in Epidermal Growth Factor Receptor (EGFR) activation in cervical cancer. However, mutations in the activation of the EGFR pathway in this type of cancer are rare. Unlike observations in head and neck tumors, HPV-negative patients had higher EGFR amplification while HPV-positive patients exhibited lower EGFR amplification. However, the reason underlying this inverse relationship remains unknown. Some theories have sought to explain this situation, one of them being that smoking induces EGFR overexpression through a mechanism of increased hypoxia in tumor tissue [45,46,47,48].

The association in the development of SNIP and EGFR exon 20 mutations was previously reported in an in vitro model. An organotypic raft culture system was compared using nasal polyp-derived immortalized NP2 (iNP2) cells expressing EGFR exon 20 mutants or an exon 19 mutant and SIP-derived iIP4 cells harboring a P772_H773insPYNP mutation. The authors found the characteristic growth patterns of SNIP in a raft culture of iIP4 cells and in an iNP2 cell culture expressing the EGFR exon 20 duplication mutants S768_D770dup and N771_H773dup, but not the EGFR exon 19 mutant E746_A750del. Importantly, this effect was inhibited by either the PI3K or MEK inhibitor. Additionally, the authors found that migration and invasion were increased in iIP4 cells and iNP2 cells when they expressed the N771_H773dup mutant. The authors concluded that specific mutations in EGFR exon 20 play a crucial role in SNIP development, partially through hyper-activation of the PI3K/AKT and MAPK signaling pathways, a signaling pattern that is also induced by the HPV E6 oncoprotein [49].

Another study included 54 patients, in which 25 SNIP, 5 oncocytic sinonasal papillomas (OSPs), 35 SCC (23 SNIP-associated SCC and 12 de novo-SCC) mutations in 10 genes (KRAS, EGFR, BRAF, PIK3CA, NRAS, ALK, ERBB2, DDR2, RET, and MAPK21), and the presence of HPV-DNA detected via in situ hybridization (ISH) were analyzed. EGFR mutations were found in 72% of SNIPs, 30% of SNIP-associated SCCs, and 17% of de novo-SCCs. The KRAS gene alone and in combination with EGFR was found to be mutated in 16% and 4% of SNIP cases, respectively. The PIK3CA gene was found to be mutated in 4% of SNIP-associated SCCs. HR-HPV was observed in only 13% of SNIP-associated SCC and 8% of de novo-SCC patients. At a 5-year follow-up, SCC was observed in only 30% (6/20) of patients with EGFR-mutated SNIP compared to 76% (13/17) of patients with EGFR-wild-type SNIP (*p* = 0.0044). The authors concluded that a small subset of these tumors could be related to HPV infection. EGFR mutations characterized instances of SNIP with a lower risk of developing into SCC [50].

##### Association with Progression to Malignancy in the SNIP

When considering the association of HPV and the progression to malignant SNIP, the Speckled protein 100 kDa (Sp100) protein was investigated for its implications in inhibiting HPV replication and transcription in maintenance and differentiation-dependent replication [51]. One study included 10 inferior turbinates diagnosed with normal nasal mucosa as control specimens, 18 specimens diagnosed as SNIP, 15 specimens diagnosed as SNIP with dysplasia, and 7 specimens diagnosed as SNIP with malignant transformation (MT). In this study, HPV DNA was detected in 1 of 15 (6.7%) cases of SNIP with dysplasia and 3 of 7 (42.9%) SNIP cases with MT. Sp100 expression was detected using immunohistochemistry staining. Expression was found to be intense in the normal nasal group and moderately expressed in the SNIP group, with low expression in the SNIP with dysplasia group. The lowest expression was seen in the SNIP with MT group. Sp100 expression was correlated with HPV infections and SNIP with MT. However, no correlation with respect to age, gender, or smoking was observed. Subsequently, the authors hypothesized that Sp100 might play a functional role in the progression of SNIP [52].

The host’s immune system eliminates the viral infection in a short period of time. Thus, the human papillomavirus must escape this mechanism. HPV achieves this escape by reducing HLA class I human leukocytes, antigen-transporting peptides, thereby blocking the effects of type I interferon [53]. It was previously reported that patients with HPV infection presented increased concentrations of TCD4+ and TCD8, which may be a parameter of active viral elimination [54] associated with a better prognosis [52]. Most HR-HPV infections are cleared by a competent immune response, primarily mediated by CD4+ and CD8+ T lymphocytes. However, in patients with a poor immune response, viral infection persists, leading to subsequent epithelial damage [55].

The literature reports that HPV-positive tumors are associated with a greater infiltrate of CD8+ cells and B lymphocytes, with a low prevalence of M1 and M2 macrophages. No significant differences were found in the neutrophil infiltrate. On the other hand, the number of tumor B cells and stromal plasma cells is associated with greater progression-free survival [55].

Saloura et al. reported a high infiltrate of CD8+ cells and regulatory T cells in HPV-positive head and neck carcinomas compared to HPV-negative tumors, the latter presenting an infiltrate rich in M2 macrophages [56].

CD4+ lymphocyte infiltration in HPV+ tumors shows greater variability, and its prognostic impact is not as consistent as that observed for CD8+ lymphocytes. Although some studies report survival benefits, others found no significant relationship. The CD4/CD8 ratio has been explored as an alternative marker, with preliminary results suggesting a possible association with prognosis in tonsillar and base-of-tongue carcinomas. However, the evidence is limited and heterogeneous. In HPV+ laryngeal carcinomas, this ratio is usually low, without a clear correlation with clinical course [57].

In typical head and neck tumors, it was found that patients who are HPV-negative frequently have more p53 mutations and high levels of pRb (retinoblastoma protein). In contrast, HPV-positive carcinomas are associated with fewer p53 mutations and low pRb upregulation [54].

Tumor protein p53 works as a sensor of cell damage. When there is serious damage, p53 induces cell death and, when mutated, it produces mutations in germ lines. Similarly, pRb is responsible for the transcription of genes critical for the progression of cells from the G1 phase to the S phase of the cell cycle and, when mutated, causes the cell to lose control of its cell cycle [58]. In patients with head and neck cancer, the HPV E6 protein does not produce mutations in p53 [48]. Other mutations have been described, such as those found in the PIK3CA gene, reported in 30% of cases [59,60]. Greater genomic instability has also been reported as the mutation of PIK3 increases. However, in such cases, a mutation of p53 is not present, as both appear to be mutually exclusive. The association between the mutation of the p53 pathway and human papillomavirus infection has been reported in less than 1% of cases. Mutation of the PIK3 pathway and human papillomavirus infection has been reported in 58% of cases [60,61,62].

Furthermore, the presence of metalloproteinases (MMPs) has been investigated in HPV-positive cases. MMPs are proteolytic enzymes that degrade extracellular matrix (ECM) proteins, thus participating in tissue remodeling, the inflammatory process, and communication between the epithelium and the stroma. These processes are altered in SNIP and may contribute to its progression.

The presence of metalloproteinases has been investigated in HPV-positive cases. Metalloproteinase 2 (MMP-2) is primarily produced in vascular smooth muscle cells to maintain ECM homeostasis. MMP-2 is also produced by stromal cells and tumor cells to promote processes such as cell migration, invasion, and metastasis.

Metalloproteinase 9 (MMP-9) is produced in different cellular contexts, such as in response to inflammatory cytokines or oxidative stress. Epithelial cells and fibroblasts produce MMP-9, with the latter producing it in low quantities and possessing the ability to recruit MMP-9 from other sources. Importantly, tumor cells secrete MMP-9, which contributes to angiogenesis and tumor growth.

In a study with SNIP tissue, the expression of MMP2 and MMP9 was investigated, and the presence of HPV 6/11, 16/18, and 31/33 was compared. MMP-2 and 9 were observed in SNIP with moderate and severe dysplasia, specifically in carcinoma and invasive SCC, compared to control nasal mucosae. Among SNIP, HPV 6/11-positive tumors were present in 41% of cases and HPV 16/18-positive tumors present in 31%. Additionally, precancerous lesions of SNIP exhibited elevated levels of MMP-2 and 9 [63]. (see Figure 1C).

##### HPV Association with SNIP Recurrence

HPV+ tumors typically show greater infiltration of CD8+ than that in HPV− tumors. High CD8+ density is consistently associated with better overall survival, disease-free survival, and locoregional control, making it one of the most important immunological biomarkers for HPV+ oropharyngeal carcinoma. In other tumors, this effect is less consistent, although the presence of CD8+ cells is associated with a more effective antitumor immune response [64].

A previous study aimed to determine the factors associated with HPV-positive results and the recurrence of SNIP. Clinical data and fresh tissue samples were prospectively collected from 90 consecutive patients treated for SNIP, among whom 14 had recurrent SNIP. Tissue samples were analyzed for the presence of HPV and factors associated with recurrence (smoking, alcohol consumption, Krouse classification, HPV vaccination, and HPV status). Among 107 SNIP specimens, 14 (13.1%) were positive for LR-HPV and 6 (5.6%) were positive for HR-HPV. Being HPV positive was associated with an increased risk of recurrence (*p* = 0.004). Smoking was significantly associated with HPV-positive SNIP (*p* = 0.01). The recurrence rate was lower among patients with SNIP that underwent an attachment-oriented resection compared to patients treated without attachment-oriented resections (78.6% vs. 25.8%, *p* < 0.001). SNIP recurrence was highly associated with HPV-positive results and surgery without an attachment-oriented resection. Importantly Oncogenic HPV was rare in SNIP [65].

A previous systematic review and meta-analysis of 14 eligible studies including 592 patients with SNIP concluded that HPV-positive cases exhibited a significantly higher odds ratio (OR) for tumor recurrence than HPV-negative cases, suggesting a pathological role of HPV in SNIP [57]. Another systematic review and meta-analysis, aimed at determining the pooled odds ratios (ORs) and 95% confidence intervals (CI) included case–control studies reporting SNIP recurrence data and HPV status identified via polymerase chain reaction (PCR) and ISH. Twenty-five studies were identified, encompassing a total of 1116 benign SNIP tumors. A total of 267 SNIPs were HPV-positive, among which 103 were recurrent, and 849 SNIPs were HPV-negative, among which 231 were recurrent. The pooled standardized odds ratio for recurrence in HPV-positive tumors was 2.05 (95% CI: 1.31–3.19). HPV subtypes did not show statistical significance. Therefore, HPV infection may be associated with an increased risk of SNIP recurrence, regardless of HPV subtype [66].

#### 2.4.4. Epigenetic Alterations in HPV-Related SNIP

In recent years, epigenetic regulation has been shown to play a fundamental role in the mechanisms of carcinogenesis and other cellular alterations [67]. Regarding the etiology of SNIP, there is increasing evidence suggesting that HPV infection may cause epigenetic alterations that play an important role in the development of or progression to malignancy [68].

The development of SNIP is closely related to HPV infection. Specifically, HPV11 exhibits the highest expression in tissues with SNIP. High-throughput sequencing identified the histone lysine demethylase 4a (KDM4A) site as the HPV integration site in HPV-positive samples [44]. KDM4A expression promotes tumor cell proliferation, invasion, and metastasis and plays a crucial role in epigenetic regulation through the association between histone deacetylase (HDAC) and p53 [69].

In HNE-pC cells, HPV11E6/E7 was overexpressed, and the KDM4A gene was inactivated using CRISPR/Cas9 technology. The effect on cell proliferation and migration was then evaluated alongside the effect of macrophage polarization in human mononuclear leukemia cells (TPH-1). The authors found that HPV11E6/E7 overexpression significantly increased the proliferation and migration of nasal epithelial cells in addition to promoting M1 macrophage polarization. KDM4A inactivation inhibited these effects and delayed the progression of macrophages toward M1 polarization. Therefore, HPV11 (LR-HPV) may promote nasal mucosal proliferation and regulate M1 macrophage polarization through KDM4A, which could contribute to the pathogenesis of SNIP [44].

A study on SNIP samples analyzing mutations in the genes and their association with HPV found LINE-1 hypomethylation in approximately 20% of SNIP and OSP samples, 52% of SNIP squamous cell carcinomas, and 58% of de novo squamous cell carcinomas. Notably, LINE-1 methylation levels significantly decreased, transforming from papilloma (mean 57.2%) to early stage (56%) and, finally, advanced stage SCC groups (46.4%). The authors concluded that LINE-1 hypomethylation is associated with occupational exposure and could be used to identify more aggressive nasal SCC [50].

## 3. Diagnosis

The definitive diagnosis of SNIP is histopathological. However, up to 17% of biopsies may be false negatives [3], so it is important to perform a deep biopsy with sufficient tissue to reduce the risk of misdiagnosis. Within the diagnostic approach, the recommended initial imaging modality is a CT scan (GE Healthcare, Chicago, IL, USA) of the nose and paranasal sinuses, which allows for tumor staging according to its extent. The Krouse classification is the most widely used method internationally and is divided into four stages. T4 is the most advanced stage, which is associated with malignancy or intracranial extension [70]. CT is also very useful for identifying the tumor implantation site, which appears as an area of hyperostosis. CT allows for complete surgical resection with reaming of the insertion site, thereby reducing the risk of recurrence.

Magnetic resonance imaging (MRI) (Siemens Healthineers, Erlangen, Germany) is another imaging technique that enables the analysis of involved soft tissues, primarily at the T4 stage. Classic T2-weighted images show a cerebriform pattern related to epithelial invaginations [3].

## 4. Treatment

The gold standard for managing SNIP is surgical resection using an endoscopic approach, open surgery, or combined approaches. Endoscopic procedures are currently considered the preferred method, provided they are performed in selected cases where the tumor’s location and size allow for complete resection. Technology in angled instruments and endoscopes makes such procedures possible. However, there are some contraindications for the endoscopic approach, such as T4 stage tumors with intracranial or intraorbital extension or certain frontal sinus tumors that require open surgery or a combined approach. In some cases, a multidisciplinary approach involving neurosurgery or orbital surgery is also necessary [71].

A detailed pre-surgical analysis of imaging studies is crucial, as identifying the implantation site allows for a surgical plan that includes intraoperative exploration of the tumor insertion point and reaming of the underlying bone. For the identification of this site, CT has low sensitivity (approximately 50%) but high specificity, with a reported positive predictive value (PPV) of up to 100% [72].

If the implantation site is identified preoperatively and/or intraoperatively, there is a clear consensus in the literature that resection of the insertion site is essential for effective removal of the SNIP. Incomplete resection has been reported as the main risk factor for recurrence. Therefore, the surgical technique should include complete resection of the diseased mucosa and the apparently healthy mucosa underlying the insertion site, in addition to cauterization or reaming of the bone and mucosa underlying the insertion site [72].

Kim et al. performed a meta-analysis to assess tumor recurrence according to the type of surgical approach in patients with SNIP. The results suggested the use of endoscopic surgery for Krouse stages T1 or T2 and an open approach for stage T3, which presents a lower recurrence rate compared to the endoscopic approach [73].

### Follow-Up

Recurrence rates vary widely among reports, from 0% to 50%. A meta-analysis reported that the average recurrence rate, considering all stages, was 15% after a mean follow-up of 44 months [3].

Currently, the follow-up time for patients with SNIP is variable. However, most current studies recommend a minimum of 3 years—and ideally 5 years—although some authors recommend follow-ups for life [3].

Most recurrences occur within the first two years after surgery. A multicenter study showed a significant difference in the recurrence rate between patients with more than three years of follow-up compared to those with shorter follow-up periods, with recurrence rates of 26.1% and 8.5%, respectively [74]. Furthermore, the risk of malignant transformation is a determining factor for maintaining long-term patient follow-up. One of the most widely accepted follow-up schedules involves visits every three to four months during the first year, followed by every six months during the second year, and every six to twelve months thereafter. A clinical evaluation with nasal endoscopy is performed at each visit. If recurrence is suspected, imaging studies, including CT and MRI, are ordered [3].

The treatment for recurrence involves complete surgical resection, aiming to identify the anatomical site of implantation to manage the mucosa and bone in that area by reaming out the hyperostosis [3].

Finally, factors that cause chronic inflammation, such as smoking or exposure to certain chemical solvents, must be eliminated while the HPV genome is cleared via the patient’s immune response, with the patient receiving appropriate treatment based on a medical evaluation. Although risk factors associated with the development of SNIP have been evaluated in various studies, the populations differed in terms of their sociodemographic characteristics and exposure history, as well as the genotypes characterized in the tissue samples and the duration of the infection, among other factors. Additionally, it may not be a single factor but, rather, a combination of factors that creates a microenvironment favorable for HPV replication, leading to increases in the number of genome copies in the progression to malignancy and persistence that leads to recurrence.

## 5. Conclusions

The development of SNIP has been associated with environmental factors (primarily a history of smoking) and chronic inflammation that damages the respiratory mucosa, initially through microlesions that allow HPV to enter. However, this review has limitations, as the described publications varied in terms of the sociodemographic characteristics of their study populations, such as age, sex, and geographic location, as well as in sample size and virus detection methodology. Nevertheless, HPV has been found in approximately 40% of samples that progress to malignancy and exhibit a higher recurrence rate. The microenvironmental characteristics of HPV-positive samples that progress to malignancy include a decrease in HLA class I human leukocytes, interferon blockade, an increase in CD4+ T cells and an even greater increase in CD8+ T cells, a decrease in M1 and M2 macrophages, and the presence of metalloproteinases 2 and 9, among others. Therefore, studies on the SNIP microenvironment should be performed before and after resection, including HPV testing, without high- or low-risk classification being used as a determining factor, with a follow-up of at least five years.

## Figures and Tables

**Figure 1 ijms-27-00245-f001:**
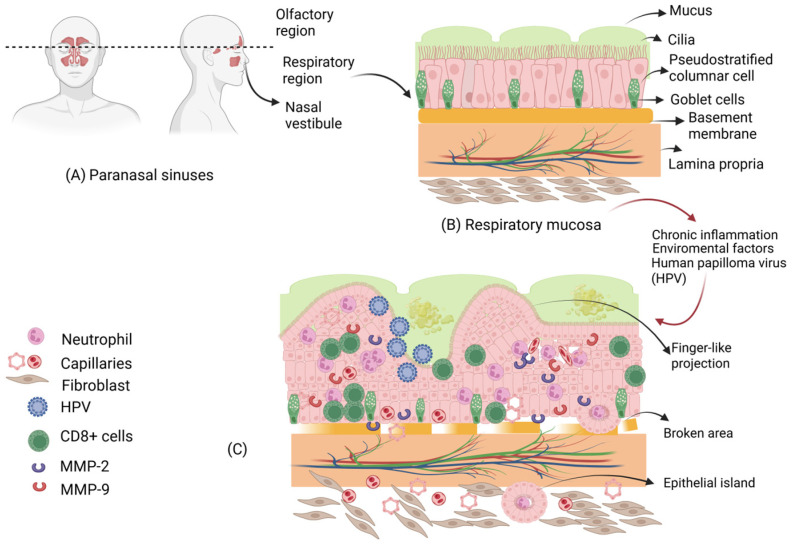
Sinonasal components. (**A**) Location of the paranasal sinuses. (**B**) Normal sinonasal mucosa showing the mucus layer, ciliated epithelium, pseudostratified epithelium, goblet cells, basement membrane, lamina propria with its blood vessels, and fibroblasts. (**C**) Findings in cases of chronic inflammation, affecting environmental factors, and human papillomavirus. Finger-like projections, loss of basement membrane continuity, and cellular microenvironment are shown. The dashed line indicates the imaginary separation between the olfactory region and the respiratory region. The red arrows indicate the transition from normal respiratory mucosa to sinonasal inverted papilloma. Created using https://BioRender.com.

## Data Availability

No new data were created or analyzed in this study. Data sharing is not applicable to this article.

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
