# Peer review of "Human Papillomavirus: Possible Mechanisms of Damage in Sinonasal Inverted Papilloma"

_ijms, 2025, doi:10.3390/ijms27010245_

Round 1

Reviewer 1 Report

Comments and Suggestions for Authors
  1. Many studies report conflicting HPV prevalence in SNIP. High-risk vs low-risk HPV distinction is important. Authors should identify this controversy in the introduction.
  2. Reduce unnecessary background detail; keep only content relevant to HPV mechanisms.
  3. Authors have to correct the description of epithelial cell types.
  4. The authors describe cells in detail but do not connect this to susceptibility to HPV infection, chronic inflammation-driven remodeling and basal cell microenvironment, where HPV is likely to replicate.
  5. Authors need to differentiate between dysplasia vs. squamous metaplasia in SNIP. It is important when discussing progression to carcinoma.
  6. Double check microvilli length (is it mm or μm).
  7. How do environmental factors, HPV infection, and EGFR mutations interact to influence SNIP development or recurrence?
  8. How do immune infiltrates (CD4+/CD8+/macrophages) relate specifically to HPV-positive SNIP versus HPV-negative SNIP?
  9. Some paragraphs mix multiple topics (e.g., HPV molecular mechanisms and EGFR mutations), making it difficult for the reader to follow.
  10. Section 2.4.4. Epigenetic alterations would benefit from explicit statements connecting epigenetic alterations to HPV infection, EGFR mutations, immune response, or environmental factors.
  11. Adding a concise, evidence-based final paragraph would enhance clarity and clinical relevance.
  12. The conclusion section does not connection is made yet to HPV mechanisms, even though the paper title and abstract emphasize HPV.
  13. Many of the studies referenced vary in sample size, geographic location, and methodology (e.g., differences in HPV detection methods, inclusion of high-risk vs. low-risk types, and genetic analyses). This heterogeneity could affect the generalizability of the conclusions; thus, acknowledging limitations helps guide readers in interpreting the strength of evidence and identifying areas for future research.

Author Response

Point-by-point response to Comments and Suggestions for Authors

  1. Many studies report conflicting HPV prevalence in SNIP. High-risk vs low-risk HPV distinction is important. Authors should identify this controversy in the introduction.

Response. Thank you for the observation, this information has been integrated into the last lines of the Introduction (underlined in yellow) and the data found are described in Section 2.4.3. Association with the human papillomavirus.

2. Reduce unnecessary background detail; keep only content relevant to HPV mechanisms.

Response. We agree, the parts describing other types of cancer associated with HPV have been removed; in epigenetics, only the relevant information associated with HPV has been described, and the rest has been removed.

3. Authors have to correct the description of epithelial cell types.

Response. The cell types have been described according to the reported histology, starting with the nasal vestibule. Additionally, the figure was corrected.

4. The authors describe cells in detail but do not connect this to susceptibility to HPV infection, chronic inflammation-driven remodeling and basal cell microenvironment, where HPV is likely to replicate.

Response. Thank you for the suggestion. The changes have been made to highlight the importance of basal (undifferentiated) cells and the changes in the expression of certain molecules when microlesions, inflammation, and susceptibility to papillomavirus replication are present. This information can be found in the 5th and 6th paragraphs of Section 2.4.3., and has also been added to the figure 1.

5. Authors need to differentiate between dysplasia vs. squamous metaplasia in SNIP. It is important when discussing progression to carcinoma.

Response. We have differentiated the terms. They are described in the histology section of the SNIP and, to avoid confusion, the HPV involvement section has been subdivided into development, progression to malignancy, and recurrence subsections.

6. Double check microvilli length (is it mm or μm).

Response: We appreciate the observation, we made a mistake and it has been corrected to μm.

7. How do environmental factors, HPV infection, and EGFR mutations interact to influence SNIP development or recurrence?

Response: The association of SNIP with HPV has been separated into three sections—development, progression to malignancy, and recurrence—for better explanation.

In the case of EGFR, it participates in the development of SNIP in HPV-positive patients, although EGFR amplification is lower and the reason for this inverse relationship is unknown. However, there are theories that attempt to explain this situation; one of them is that smoking induces EGFR overexpression through a mechanism of increased hypoxia in tumor tissue with a history of smoking, where this factor is increased. Additionally, in a 5-year follow-up study, HPV-positive tissue and mutated and wild-type EGFR were found in SNIP tissue (see description in the manuscript).

Regarding SNIP recurrence, the work of Viitasalo S., 2022, has been described, where they found that recurrence was strongly associated with HPV positivity and non-resection surgery focused on the insertion site.

8. How do immune infiltrates (CD4+/CD8+/macrophages) relate specifically to HPV-positive SNIP versus HPV-negative SNIP?

Response: Immune infiltrates are not specific; the studies described have found a greater quantity and activity of CD8+ and also CD4+ cells in HPV+ samples. The explanation can be found starting in the second paragraph of the section on association with progression to malignancy.

Regarding macrophages towards M1, in vitro work is described using Human Nasal Epithelial Cells (HNEpC) infected with the lentivirus HPV11E6/E7, which are inoculated into female BALB/c nude mice. The tumor is then monitored, and M1 macrophage infiltration is found. The explanation can be found in the third paragraph of the section on HPV association and the development of SNIP.

9. Some paragraphs mix multiple topics (e.g., HPV molecular mechanisms and EGFR mutations), making it difficult for the reader to follow.

Response: We appreciate the feedback and have removed the sections unrelated to HPV. The relevant text has been divided into sections covering development, progression, and recurrence.

10. Section 2.4.4. Epigenetic alterations would benefit from explicit statements connecting epigenetic alterations to HPV infection, EGFR mutations, immune response, or environmental factors.

Response: I completely agree. When we re-read the document, we realized it was not clear. Therefore, it was narrowed down to HPV-related epigenetic alterations in the SNIP.

11. Adding a concise, evidence-based final paragraph would enhance clarity and clinical relevance.

Response: We appreciate the observation. The observation was added to the last paragraph of the follow-up section.

12. The conclusion section does not connection is made yet to HPV mechanisms, even though the paper title and abstract emphasize HPV.

Response: We agree with this observation, and the Conclusion has been changed to include some of the mechanisms described.

13. Many of the studies referenced vary in sample size, geographic location, and methodology (e.g., differences in HPV detection methods, inclusion of high-risk vs. low-risk types, and genetic analyses). This heterogeneity could affect the generalizability of the conclusions; thus, acknowledging limitations helps guide readers in interpreting the strength of evidence and identifying areas for future research.

Response: You are right, we have changed the Conclusion and indicated that it cannot be generalized due to the differences in the studies.

Reviewer 2 Report

Comments and Suggestions for Authors

The manuscript is difficult to follow. First part is very good, then it gets confusing in the HPV section. Would recommend re-writing, with clear statement.

Also discuss outcomes in squamous cell carcinoma- ex SIP versus squamous cell carcinoma de novo.

Author Response

Point-by-point response to Comments and Suggestions for Authors

  1. The manuscript is difficult to follow. First part is very good, then it gets confusing in the HPV section. Would recommend re-writing, with clear statement.

Response: We appreciate the observation. When we reread the manuscript, we realized how confusing the section related to HPV was. We rewrote it, organizing it into mechanisms described for the development of SNIP, mechanisms of progression to malignancy and, finally, recurrence.

  1. Also discuss outcomes in squamous cell carcinoma- ex SIP versus squamous cell carcinoma de novo.

Response: Thank you, this point has been clarified in Section 2.2. Medical Importance, and is underlined in yellow.

Round 2

Reviewer 2 Report

Comments and Suggestions for Authors

Critiques and suggestions have been appropriately addressed- thank you.